# FedDEO: Description-Enhanced One-Shot Federated Learning with Diffusion Models

## ABSTRACT

In recent years, the attention towards One-Shot Federated Learning (OSFL) has been driven by its capacity to minimize communication. With the development of the diffusion model (DM), several methods employ the DM for OSFL, utilizing model parameters, image features, or textual prompts as mediums to transfer the local client knowledge to the server. However, these mediums often require public datasets or the uniform feature extractor, significantly limiting their practicality. In this paper, we propose FedDEO, a **D**escription-**E**nhanced **O**ne-Shot **Fed**erated Learning Method with DMs, offering a novel exploration of utilizing the DM in OSFL. The core idea of our method involves training local descriptions on the clients, serving as the medium to transfer the knowledge of the distributed clients to the server. Firstly, we train local descriptions on the client data to capture the characteristics of client distributions, which are then uploaded to the server. On the server, the descriptions are used as conditions to guide the DM in generating synthetic datasets that comply with the distributions of various clients, enabling the training of the aggregated model. Theoretical analyses and sufficient quantitation and visualization experiments on three large-scale real-world datasets demonstrate that through the training of local descriptions, the server is capable of generating synthetic datasets with high quality and diversity. Consequently, with advantages in communication and privacy protection, the aggregated model outperforms compared FL or diffusion-based OSFL methods and, on some clients, outperforms the performance ceiling of centralized training.

## CCS CONCEPTS

• **Computing methodologies** → **Computer vision**.

## KEYWORDS

One-Shot Federated Learning, Diffusion Model

## 1 INTRODUCTION

Federated Learning (FL) [22] has attracted increasing attention due to its capability to enable collaborative training across multiple clients while ensuring the protection of user data within local environments. In contrast to centralized training that involves the direct uploading of client data to the server, FL fundamentally operates

as a distinctive form of knowledge transfer. In FL, all participating clients transfer knowledge in their local data to the aggregated model without sharing the raw local data. Previous extensive works have aimed to enhance the efficiency of this knowledge transfer, leading to the development of One-Shot Federated Learning (OSFL).

In OSFL, clients are tasked with using some **mediums** to transfer the information about the local distributions to the server within a single communication round. The traditional OSFL methods primarily utilize two kinds of mediums. Firstly, model parameters serve as the primary medium. Parameters of generative models trained on clients are uploaded to the server for generating pseudo-samples to train the aggregated model [8, 43]. Additionally, distilled client data [47] can also serve as the medium and be uploaded to the server for the training of the aggregated model. Indeed, the practical deployment of these mediums is challenging due to the difficulty in training generative models and the privacy concerns and communication associated with uploading data.

In recent years, the development of diffusion models (DMs) has brought new opportunities for OSFL. There are powerful pre-trained DMs that learn vast knowledge from large-scale datasets, allowing them to generate data complying with most of common distributions as long as suitable condition is provided. Moreover, impressively, due to the knowledge from the pre-trained DMs, the trained aggregated model has the potential to outperform the performance ceiling in traditional FL methods, which involves uploading all client data to the server for the centralized training of the aggregated model. This brings significant changes for OSFL. A small amount of descriptions of the client data distribution are sufficient to serve as the medium for transferring client knowledge and guiding the conditional generation on the server.

Some works has already begun to explore how to describe the distribution of client data. FedDISC [41] utilizes client image features. FGL [44] uses textual descriptions of the client data generated by BLIPv2 [14]. FedCADO [42] employs local classifiers trained on clients. All these methods have demonstrated the significant potential of DMs in OSFL. But FedDISC requires the sharing of pre-trained feature extractors between the server and clients, limiting its practicality and flexibility. The text descriptions uploaded by FGL may involve more direct privacy risks, and the classifiers uploaded by FedCADO typically only provide relatively vague guidance. We aim to find a more **flexible**, **practical**, **accurate**, and **privacy-protecting** description of client data distribution to serve as the medium for transferring knowledge from the clients. A natural solution arises: training the local descriptions of the client data and utilizing them for guiding the image generation on the server.

Based on this idea, in this paper, we introduce a **D**escription-**E**nhanced **O**ne-shot **Fed**erated learning method with diffusion models, **FedDEO**, where we employ a learnable vector as the description for the client distribution to be the medium of client knowledge. In brief, our method consists of two main components: **client description training** and **server image generation**. On the clients,

we employ the noise-predicting ability of the DMs to train the descriptions of the client data and embeds personalized distribution information into the descriptions, which are subsequently uploaded to the server. On the server, we utilize the descriptions uploaded by each client to guide the server's pre-trained DM in generating a synthetic dataset that complies with the clients' distributions, and use the synthetic dataset to train the aggregated model. Training the aggregated model based on synthetic datasets means we can flexibly select the model structure. The guidance provided by descriptions trained by pre-trained DMs is more accurate, improving the quality of diffusion generation. These descriptions are merely vectors, which significantly reduces the communication of the clients, enhancing the practicality of the method. Due to the randomness of the diffusion generation process, it is almost impossible to directly or indirectly obtain sensitive client privacy information through the descriptions or the generated samples. The description serves as the medium that perfectly meets our requirements for flexibility, practicality, accuracy, and privacy protection.

To demonstrate the validity of FedDEO, on one hand, we conduct thorough theoretical analyses. We prove the boundedness of the Kullback-Leibler (KL) divergence between the distribution of the synthetic data and the distribution of client local data. This provides theoretical assurance for the server's generation of synthetic data that complies with the client distributions. On the other hand, to further validate the performance of our method, we conduct extensive quantitation and visualization experiments on three large-scale real-world datasets, DomainNet [25], OpenImage [13], and NICO++ [46]. The visualization experiments demonstrate that our method is capable of generating a synthetic dataset that complies with the distribution of each client's data. Moreover, the quality and diversity of the synthetic dataset are comparable with the original client datasets. In our quantitation experiments, we explore scenarios with varying numbers of clients, skews in feature distribution and label distribution among clients, as well as the utilization of different pre-trained DMs. These quantitation experiments also demonstrate that the aggregated model trained by FedDEO outperforms other compared methods, and notably, on NICO++, it outperforms the ceiling performance of traditional FL frameworks that involve uploading all client data to the server. Additionally, we conduct comprehensive quantitation and visualization experiments to demonstrate FedDEO's performance in terms of privacy protection, communication, and computational efficiency. All these findings demonstrate the performance of our method.

In summary, the contributions of this paper are as follows:

- We propose FedDEO, realizing one-shot federated learning in realistic scenarios and further exploring the potential of utilizing diffusion models in federated learning.
- We introduce the trained descriptions of the client distributions as the novel medium for transferring the client knowledge, which perfectly meets the practical requirements for flexibility, practicality, accuracy, and privacy protection.
- We conduct theoretical analyses, demonstrating the validity of generating synthetic data complies with client data distributions conditioned on the local descriptions.
- Extensive quantitative and visual experiments are conducted to validate the performance of our method, demonstrating

that FedDEO can generate high-quality synthetic datasets and train the aggregated model that outperforms other compared methods, even the performance ceiling of centralized training.

## 2 RELATED WORK

### 2.1 Diffusion Model

The DM is first introduced in [33]. The overall framework of current DMs occurs in [9]. Subsequent sampling methods, including DDIM[34] and PNDM [19], further improved the quality and efficiency of DMs' generation. [12] and [3] demonstrate excellent generation results on real images. The stable diffusion based on LDM [29], which pre-trains on large-scale datasets, has ignited a trend in the AIGC (Artificial Intelligence Generated Content). One notable characteristic of stable diffusion is its conditional generation capability. Given suitable conditions as guidance, such as image [31, 37, 38, 45], text [11, 24, 26, 32] or the graient of loss function [3, 5, 39], stable diffusion can generate images complying with almost any distribution we encounter in our daily lives. The generated images exhibit remarkable quality and diversity. There are also some methods [6, 7, 23, 30] that fine-tune the input conditions of stable diffusion using specific datasets. These methods enable the conditions to learn the distribution of the used dataset and generate synthetic data that conforms to the specific distribution. Additionally, DMs possess the ability for compositional generation [4, 20], allowing them to handle scenarios where multiple conditions are simultaneously provided as guidance. These lead us to consider the application of these powerful pre-trained DMs in federated learning. If we can obtain guidance from clients to guide the pre-trained DM on the server, we can generate data that comply with client distributions and address the challenges of OSFL.

### 2.2 One-Shot Federated Learning

The high communication cost has been one of the major challenges faced by the classic FedAvg [22] algorithm in federated learning since its proposal. Some efforts [1, 10, 15] have addressed the non-iid problem from the perspective of optimizing algorithms to improve communication efficiency. Others [2, 16, 21] have focused on enhancing the practicality of FedAvg in non-iid scenarios from a personalized perspective. However, these efforts still encounter high communication costs. Recently, some works [8, 35, 41, 43, 47] have focused on one-shot federated learning (OSFL), considering federated learning within the constraints of a single communication round. This setting differs from standard federated learning in that OSFL allows clients to train their local models to converge in a single round and then send them to the server for aggregation.

The essence of OSFL is that all clients transfer their local knowledge to the server for aggregation through some medium. One category of work uses model parameters as the medium, such as DENSE [43], which collects classifiers trained by clients and uses them to train a generator on the server, followed by using the generator to produce pseudo-samples for knowledge distillation. FedCVAE [8] involves training conditional-VAE on clients and sending the decoders to the server, where the server generates pseudo-samples through the decoders. Another category of work uses distilled datasets as the medium; DOSFL [47] distills

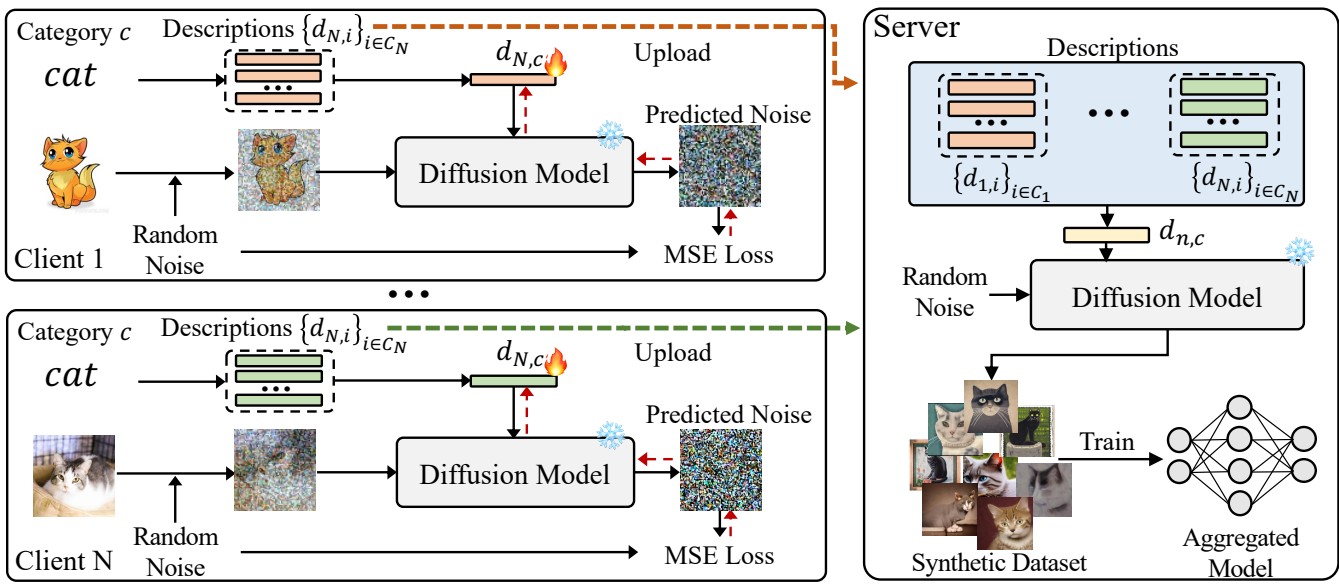

**Figure 1: The overall framework of FedDEO, including two main parts: Client Description Training and Server Image Generation. Firstly, each client trains local descriptions based on the client data and the pre-trained diffusion model, then uploads them to the server. Guided by these descriptions, the server utilizes the diffusion model to generate the synthetic dataset that complies with the various client distributions and trains the aggregated model.**

privacy data on clients and sends the distilled data to the server for aggregation. Additionally, recent efforts have combined pre-trained DMs with OSFL. FedDISC [41] uses the data features as the medium, leveraging client data features as conditions for a pre-trained DM to generate pseudo-samples. Similarly, FGL [44] uses text prompts as the medium, extracting text prompts of the local data using BLIPv2 [14] and sending them to the server as conditions for the DM. FedCADO [42] utilizes classifiers trained on clients as a medium, using the classifier-guided diffusion to generate the synthetic dataset and train the aggregated model. In contrast to these works, we train descriptions on clients as the medium. Compared to FedDISC, FedCADO and FGL, local descriptions can better capture information about the distribution of client local data and provide suitable guidance to the server to generate pseudo-samples that better comply with the client's data distribution.

## 3 METHOD

In this section, we elaborate on our method in two parts, including Client Description Training and Server Image Generation, followed by the theoretical analyses about the distributions of the synthetic data and the client local data. The overall framework of FedDEO is illustrated in Figure 1 and the pseudocode of FedDEO is provided in the supplementary materials.

### 3.1 Preliminaries

**Notations and Objectives.** In this paper, we aim to address the standard one-shot federated learning setting. Assuming we have $N$ client datasets $\mathcal{D}_n, n = 1, \ldots, N$, and collectively, these datasets

encompass a total of $M$ categories. The objective of OSFL is to obtain a global aggregated model $\mathbf{w}_g$ within a single communication round, minimizing the global objective function:

$$F(\mathbf{w}) = \frac{1}{N} \sum_{n=1}^{N} \mathbb{E}_{\mathbf{x}, y \sim \mathcal{D}_n} [\mathcal{L}_n(\mathbf{x}, y, \mathbf{w})], \tag{1}$$

where $y \in C_n$, and $C_n$ represents the set of categories owned by each client and is a subset of $\{1, \ldots, M\}$.

From this objective function, it is evident that our goal is to train an aggregated model that adapts to all client distributions and exhibits excellent classification performance on the data from each client. We also assess the model performance in subsequent experimental sections according to this objective.

**An Assumption about Diffusion Model.** The pre-trained DM is a key component of our method. As stated in the Introduction, our motivation for using pre-trained DM lies in its ability to generate synthetic data that complies with the client distributions. This is because the data used in the pre-training process of these DMs covers almost all common distributions. This motivation implies an assumption: the DMs we use have been sufficiently pre-trained to cover the data distribution of the clients. Therefore, regarding the data distribution $p_n(\mathbf{x})$ of the client's local dataset $\mathcal{D}_n$ and the data distribution $p_{\epsilon_\theta}(\mathbf{x})$ that the DMs $\epsilon_\theta$ can generate, we can make the following assumption:

**Assumption 1** *There exists $\lambda > 0$ such that the Kullback-Leibler divergence from $p_n(\mathbf{x})$ to $p_{\epsilon_\theta}(\mathbf{x})$ is bounded above by $\lambda$:*

$$KL(p_{\epsilon_\theta}(\mathbf{x}) \| p_u(\mathbf{x})) < \lambda \tag{2}$$

It's evident that this assumption is flexible. We don't strictly restrict the DM's distribution $p_{\epsilon_\theta}(\mathbf{x})$ to entirely cover the client distributions $p_n(\mathbf{x})$. Instead, we only require some overlap of the distributions, as totally non-overlap distributions is unreasonable. Even if the clients focus on some professional fields, such as medical images, it is entirely feasible to train the specialized DMs on the server. Therefore, this is a completely reasonable assumption made based on a comprehensive consideration of practical scenarios, and it is also the motivation of our method.

### 3.2 Client Description Training

Firstly, the server distributes the pre-trained DM $\epsilon_\theta$ to the clients. Based on $\epsilon_\theta$, the clients train the local descriptions capturing the characteristics of the client's data distribution. We first initialize the local descriptions. Then, we fix the parameters of the pre-trained DMs and train the local descriptions on the client's local data.

**Description Initialization.** For each category $c$ within the client $n$, we define a vector $\mathbf{d}_{n,c}, c \in C_n$ as the description of the distribution of this category. To ease the training process and consequently lessen the computation on clients, we initialize each description using the text features $f_c$ of the name of category $c$ extracted by CLIP [28], which is sent by the server. This initialization ensures that the descriptions possess the capability to guide the DM in generating images of the correct category from the beginning.

**Image Noising.** During training, for each sample $\mathbf{x}_0 \in D_n$, we start by sampling a random Gaussian noise $\epsilon$ from the standard Gaussian distribution $\mathcal{N}(0, \mathcal{I})$. We randomly sample a timestep $t$ from $\{0, ..., T\}$, where $T$ is the maximum timestep defined by the pre-trained DM. Based on $t$, we adjust the intensity of the sampled Gaussian noise and obtain the noised sample $\mathbf{x}_t$ as follows:

$$\mathbf{x}_t = \sqrt{a_t}\mathbf{x}_0 + \sqrt{1 - a_t}\epsilon, \tag{3}$$

where $a_t$ is the variance schedule defined by the pre-trained DM and changes with the timestep $t$.

**Description Training.** After obtaining $\mathbf{x}_t$, we use the description $\mathbf{d}_{n,c}$ as a condition for $\epsilon_\theta$ to predict the noise $\epsilon$ within $\mathbf{x}_t$ and employ Mean Squared Error (MSE) Loss to compute the difference between the predicted noise $\epsilon_\theta(\mathbf{x}_t, t|\mathbf{d}_{n,c})$ and the actual added noise $\epsilon$. During this process, we fix all parameters of the pre-trained DM $\epsilon_\theta$ and solely use the backpropagation to train the description $\mathbf{d}_{n,c}$. Therefore, the loss function used during the training of the description is as follows:

$$\mathcal{L}(\mathbf{x}_t, \mathbf{d}_{n,c}, t) = \mathcal{L}_{MSE}(\epsilon, \epsilon_\theta(\mathbf{x}_t, t|\mathbf{d}_{n,c})) \tag{4}$$

After $S$ epochs of training, the description can capture the characteristic of the client distributions and become proficient in guiding the pre-trained DM to effectively denoise the noise-added images. Therefore, during server image generation, with these uploaded descriptions, the DM can denoise the randomly sampled initial noise as $\mathbf{x}_T$ and generates high-quality synthetic images that comply with the client distributions guided by the trained descriptions.

### 3.3 Server Image Generation

Upon receiving the local descriptions $\{\mathbf{d}_{n,c}\}$ from client $n$, the server utilizes these local descriptions to guide the pre-trained DM in generating samples that comply with the data distribution of the client $n$ and trains the aggregated model.

**Image Generation.** Firstly, we sample the random initial noise $\mathbf{x}_T$ from $\mathcal{N}(0, \mathcal{I})$ as the start for denoising. Through multiple iterations of the timestep $t = \{T, ..., 0\}$, we denoise $\mathbf{x}_T$ to obtain the realistic sample $\mathbf{x}_0$. Specifically, to further ensure that the generated images possess accurate semantic information for the specified category, we use compositional diffusion using the text feature $f_c$ of the specified category $c$ and the local description $\mathbf{d}_{n,c}$ capturing the personalized distribution of category $c$ on client $n$. At each timestep $t$, we employ both $f_c$ and $\mathbf{d}_{n,c}$ as conditions for the compositional DM. These two conditions are separately inputted into the DM for noise prediction. And the predicted noises are accumulated, resulting in the final predicted noise as follows:

$$\hat{\epsilon}_\theta(\mathbf{x}_t, t|f_c, \mathbf{d}_{n,c}) = \epsilon_\theta(\mathbf{x}_t, t|\mathbf{d}_{n,c}) + \epsilon_\theta(\mathbf{x}_t, t|f_c) \tag{5}$$

After obtaining $\hat{\epsilon}_\theta(\mathbf{x}_t|f_c, \mathbf{d}_{n,c})$, we denoise the current timestep's sample $\mathbf{x}_t$ to $\mathbf{x}_{t-1}$ according to the following formula:

$$\mathbf{x}_{t-1} = \sqrt{\alpha_{t-1}}\left(\frac{\mathbf{x}_t - \sqrt{1 - \alpha_t}\hat{\epsilon}_\theta(\mathbf{x}_t|f_c, \mathbf{d}_{n,c})}{\sqrt{\alpha_t}}\right)$$
$$+ \sqrt{1 - \alpha_{t-1} - \sigma_t^2}\hat{\epsilon}_\theta(\mathbf{x}_t|f_c, \mathbf{d}_{n,c}) + \sigma_t \boldsymbol{\varepsilon}_t, \tag{6}$$

where $\alpha_t$ and $\sigma_t$ are pre-defined by the pre-trained DM, and $\boldsymbol{\varepsilon}_t$ is random noise sampled from $\mathcal{N}(0, \mathcal{I})$ at each timestep $t$. After multiple iterations, the randomly sampled initial noise $\mathbf{x}_T$ is denoised into the realistic image $\mathbf{x}_0$. We can define $\mathbf{x}_0$ as $\hat{\mathbf{x}}_i^{n,c}$ and incorporate it into the synthetic dataset $\{\hat{\mathbf{x}}_i^{n,c}\}, i = \{1, ..., R\}$, where $n$ and $c$ is the client index and the category. And $R$ represents the number of images generated for each category of each client, which is set to 30 in most of our experiments.

**Aggregated Model Training.** After multiple generations, we obtain the synthetic dataset $\{\hat{\mathbf{x}}_i^{n,c}\}$. Each image in $\{\hat{\mathbf{x}}_i^{n,c}\}$ has its pseudo-label $c$, thus allowing us to directly train the aggregated model $\mathbf{w}_g$ using cross-entropy loss:

$$\mathcal{L}_{agg}(\hat{\mathbf{x}}_i^{n,c}, c) = \mathcal{L}_{CE}(\hat{\mathbf{x}}_i^{n,c}, y_i^k, \mathbf{w}_g) \tag{7}$$

According to the loss function in Eq. 7, we train until convergence to obtain the final aggregated model. A question arises: How is the performance of the aggregated model we trained? Has it learned the local knowledge from the clients? We conduct the subsequent theoretical analysis as well as extensive quantitation and visualization experiments to answer this question.

### 3.4 Theoretical Analysis

From Eq. 7, it can be seen that the performance of the trained aggregated model is entirely determined by the quality of the synthetic dataset. As mentioned in the Introduction, the performance of the aggregated model trained by FedDEO has the potential to surpass the performance ceiling of centralized training, involving uploading all client local data into the server and train the aggregated model. Therefore, we use the client local datasets to evaluate the quality of the synthetic dataset. The comparison between the distribution of synthetic data and the distribution of client local data is necessary. Based on Assumption 1, we have the following theorem:

**Theorem 1** *For the distribution of client data $p_n(\mathbf{x})$ and the conditional distribution $p_{\epsilon_\theta}(\mathbf{x}|\mathbf{d})$ of the DM $\epsilon_\theta$ conditioned the description $\mathbf{d}$ trained on the clients, we have:*

| | OpenImage | | | | | | | DomainNet | | | | | | |
|---|---|---|---|---|---|---|---|---|---|---|---|---|---|---|
| | client0 | client1 | client2 | client3 | client4 | client5 | average | clipart | infograph | painting | quickdraw | real | sketch | average |
| *Ceiling* | *49.88* | *50.56* | *57.89* | *59.96* | *66.53* | *51.38* | *56.03* | *47.48* | *19.64* | *45.24* | *12.31* | *59.79* | *42.35* | *36.89* |
| FedAvg | 42.48 | 47.24 | 47.01 | 51.28 | 61.87 | 45.47 | 49.22 | 37.96 | 12.55 | 34.41 | 5.93 | 51.33 | 32.37 | 29.09 |
| FedDF | 43.26 | 44.98 | 52.54 | 56.71 | 62.89 | 48.37 | 51.45 | 38.09 | 13.68 | 35.48 | 7.32 | 53.83 | 34.69 | 30.51 |
| FedProx | 44.99 | 48.83 | 49.25 | 56.68 | 61.23 | 46.07 | 51.17 | 38.24 | 12.46 | 37.29 | 6.26 | 54.88 | 35.76 | 30.81 |
| FedDyn | 46.93 | 46.08 | 52.44 | 54.67 | 62.84 | 47.73 | 51.78 | 40.12 | 14.77 | 36.59 | 7.73 | 54.85 | 34.81 | 31.47 |
| Prompts Only | 32.91 | 33.24 | 41.72 | 45.02 | 49.85 | 35.97 | 39.78 | 31.80 | 11.61 | 31.14 | 4.13 | **61.53** | 31.44 | 28.60 |
| FedDISC | 47.42 | 49.65 | 54.73 | 53.41 | 60.74 | 52.81 | 53.12 | 43.89 | 14.84 | 38.38 | 8.35 | 56.19 | 36.82 | 33.07 |
| FGL | 48.21 | 49.16 | 54.98 | 55.47 | 63.14 | 49.32 | 53.38 | 41.81 | 15.30 | 40.67 | 8.79 | 57.58 | 39.54 | 33.94 |
| FedCADO | 48.99 | 51.66 | 55.59 | 52.80 | 62.41 | **58.86** | 55.05 | 44.25 | 17.51 | 38.74 | 9.43 | 57.31 | 38.44 | 34.28 |
| FedDEO | **51.08** | **52.53** | **61.22** | **62.18** | **67.31** | 56.68 | **58.50** | **46.77** | **18.28** | **43.97** | **10.73** | 60.64 | **41.45** | **36.08** |
| | Common NICO++ | | | | | | | Unique NICO++ | | | | | | |
| | autumn | dim | grass | outdoor | rock | water | average | client0 | client1 | client2 | client3 | client4 | client5 | average |
| *Ceiling* | *62.66* | *54.07* | *64.89* | *63.04* | *61.08* | *54.63* | *60.06* | *79.16* | *81.51* | *76.04* | *72.91* | *79.16* | *79.29* | *78.01* |
| FedAvg | 52.51 | 40.45 | 57.21 | 51.59 | 49.31 | 43.56 | 49.11 | 67.31 | 74.73 | 69.01 | 64.37 | 73.07 | 67.87 | 69.39 |
| FedDF | 50.44 | 39.62 | 57.42 | 52.91 | 51.61 | 44.76 | 49.46 | 69.79 | 78.90 | 69.53 | 66.01 | 74.86 | 70.80 | 71.64 |
| FedProx | 53.49 | 42.41 | 58.84 | 53.08 | 53.67 | 45.42 | 51.15 | 70.46 | 75.30 | 70.87 | 67.67 | 72.84 | 71.51 | 71.44 |
| FedDyn | 54.38 | 43.20 | 57.56 | 52.63 | 52.86 | 46.76 | 51.23 | 71.23 | 74.98 | 69.68 | 68.13 | 73.63 | 70.61 | 71.37 |
| Prompts Only | 50.49 | 38.10 | 54.53 | 49.39 | 49.12 | 41.58 | 47.20 | 69.79 | 69.14 | 69.32 | 59.89 | 67.83 | 66.42 | 67.06 |
| FedDISC | 56.82 | 51.43 | 59.45 | 56.17 | 52.32 | 45.64 | 53.64 | 74.32 | 73.47 | 71.25 | 66.79 | 75.28 | 70.06 | 71.86 |
| FedCADO | 54.63 | 49.21 | 58.13 | 54.75 | 54.64 | 47.03 | 53.06 | 75.13 | 73.30 | 70.31 | 68.88 | 73.60 | 72.51 | 72.28 |
| FGL | 57.25 | 49.35 | 61.81 | 58.42 | 54.29 | 47.62 | 54.79 | 74.62 | 79.43 | 71.26 | 68.65 | 76.37 | 74.31 | 74.10 |
| FedDEO | **71.03** | **58.02** | **73.33** | **68.53** | **68.16** | **63.04** | **67.01** | **81.25** | **86.19** | **82.94** | **79.94** | **83.85** | **80.27** | **82.40** |

**Table 1: The performances of the compared methods on OpenImage, DomainNet, and NICO++ under the non-IID feature distribution skew, where the italicized texts represent the performance ceiling of centralized training used as a reference, and bold texts represent the best performance of the compared methods.**

$$KL(p_n(\mathbf{x}) \| p_{\epsilon_\theta}(\mathbf{x}|\mathbf{d})) = \int p_n(\mathbf{x}) \log \frac{p_n(\mathbf{x})p_{\epsilon_\theta}(\mathbf{d})}{p_{\epsilon_\theta}(\mathbf{d}|\mathbf{x})p_{\epsilon_\theta}(\mathbf{x})} d\mathbf{x}$$

$$< \lambda + \mathbb{E}(\log p_{\epsilon_\theta}(\mathbf{d})) - \int p_n(\mathbf{x}) \log p_{\epsilon_\theta}(\mathbf{d}|\mathbf{x}) d\mathbf{x} \quad (8)$$

The proof of Theorem 1 is detailed in the supplementary materials. From it, we can observe that the KL divergence between the conditional distribution $p_{\epsilon_\theta}(\mathbf{x}|\mathbf{d})$, which is also the distribution of the synthetic data, and the distribution of client's local data $p_n(\mathbf{x})$ is bounded above. Furthermore, we can split this upper bound into three terms: $\lambda$, $\mathbb{E}(\log p_{\epsilon_\theta}(\mathbf{d}))$, and $-\int p_n(\mathbf{x}) \log p_{\epsilon_\theta}(\mathbf{d}|\mathbf{x}) d\mathbf{x}$. $\mathbb{E}(\log p_{\epsilon_\theta}(\mathbf{d}))$ is a constant independent of the sample $\mathbf{x}$. $\lambda$ is defined in Assumption 1 and represents the upper bound of the KL divergence between $p_n(\mathbf{x})$ and the unconditional distribution of the DM $p_{\epsilon_\theta}(\mathbf{x})$, which means the overlap between the distribution of the client local data and the pre-trained data of the DM. Meanwhile, $\int p_n(\mathbf{x}) \log p_{\epsilon_\theta}(\mathbf{d}|\mathbf{x}) d\mathbf{x}$ is the log-likelihood between the description $\mathbf{d}$ and the client distribution, representing the information of the client distribution contained by the trained description, which has been maximized during the training process of the descriptions.

From the above analysis, we have two conclusions: 1) the quality of the synthetic dataset is related to the used pre-trained DM, which aligns well with the intuition. 2) locally trained descriptions can indeed learn information about the client local distribution and effectively guide the DM to generate high-quality synthetic datasets. These conclusions regarding the quality of the synthetic dataset, and the performance of the aggregated model are further demonstrated in the subsequent experimental section.

## 4 EXPERIMENTS

### 4.1 Experimental Settings

**Dataset.** We conduct experiments on three datasets: **Domain-Net** [25], **OpenImage** [13] and **NICO++** [46]. We use DomainNet to simulate style differences within the same category, OpenImage to simulate subcategories' differences within super-categories, and NICO++ to simulate differences in background and specific object attributes. All datasets consist of large-scale real-world images with the resolution of 224x224 pixels. **DomainNet** comprises six domains: *clipart, infograph, painting, quickdraw, real*, and *sketch*. Each domain has 345 categories. Following the partition in FedDISC [41], we select 20 super-categories of OpenImage with 6 subcategories in each super-category according to the hierarchy of categories provided by OpenImage. **NICO++** involves 60 categories, with each category having six common domains shared across categories (autumn, dim, grass, outdoor, rock, water) and six unique domains specific to each category. These two scenarios are respectively referred to as the **Unique NICO++** (NICO++_U) and **Common NICO++** (NICO++_C) datasets. It is worth noticing that despite each data domain having its own textual description, only the category names are used as textual information for all generations, which is more practical. For more detailed information regarding the dataset, please refer to the supplementary materials.

**Client Partition.** To validate FedDEO under two scenarios of non-IID data distributions: Feature Distribution Skew and Label Distribution Skew, we partition the clients differently. To simulate Feature Distribution Skew, we set up six clients for each dataset. Each client possesses the same set of categories but with completely different data domains. To simulate Label Distribution Skew, we

divide the 60 categories of Common NICO++ and Unique NICO++ into six clients. Each client possesses all six data domains of 10 different categories. For all client partitions, there is no overlap between the data of different clients. Due to space limitations, for detailed client partition, including the number of images in each client dataset and experiments related to the total number of clients, please refer to the supplementary materials.

**Compared Methods.** We compare our method with nine other methods, which can be divided into three major categories: 1) **Ceiling.** The performance ceiling of traditional FL methods is centralized training, involving the uploading of all client local data for the training of the aggregated model. 2) Traditional FL methods with multiple rounds of communications: **FedAvg [22], FedDF [18], FedProx [15], FedDyn [1]**. All of them have 20 rounds of communications. Following standard experimental settings, each round involves one epoch of training on each client. And we use ImageNet as the additional public data for distillation in FedDF. 3) Diffusion-based OSFL methods: **FedDISC [41], FedCADO [42], FGL [44]** and **Prompts Only**. Although FedDISC is designed for semi-supervised FL scenarios, we remove the pseudo labeling process of FedDISC and directly utilize the true labels of client images. Another point to notice is the **Prompts Only**, where the server does not use the uploaded local descriptions from clients at all but only uses the text prompts of category names in the server image generation. It's worth noticing that some mentioned OSFL methods such as DENSE[43] and FedCVAE [8] is not used as the compared methods because of the difficulty in training generative models until convergence in the high-resolution realistic image scenarios we selected. For more details about our experimental settings and implementation, please refer to the supplementary materials.

## 4.2 Main Results

In Table 1 and Table 2, we present the performance of our method under two non-IID scenarios, feature distribution skew and label distribution skew. Several observations can be made:

- Our method demonstrates significantly superior performance on all used datasets, surpassing other compared methods and even the performance ceiling on OpenImage and NICO++, demonstrating the potential of utilizing DMs in FL.
- The reason for the better performance on OpenImage and NICO++ lies in the fact that these two datasets primarily consist of realistic images, and their distributions are closer to the distributions of the used DM. This also corroborates our theoretical analysis and the better performance on the real domain of DomainNet.
- Compared to Prompts Only, without the guidance of the trained local descriptions, the DM tends to generate more realistic images, with the specific style or subcategory of the generated images being entirely randomly determined, making it challenging to adapt to the personalized local distribution of each client except for the real domain of DoomainNet.
- Compared to other diffusion-based OSFL methods, FedDEO demonstrates more stable results, indicating that the local descriptions sufficiently trained on the clients can provide more precise guidance for the generation, thereby producing higher-quality synthetic datasets.

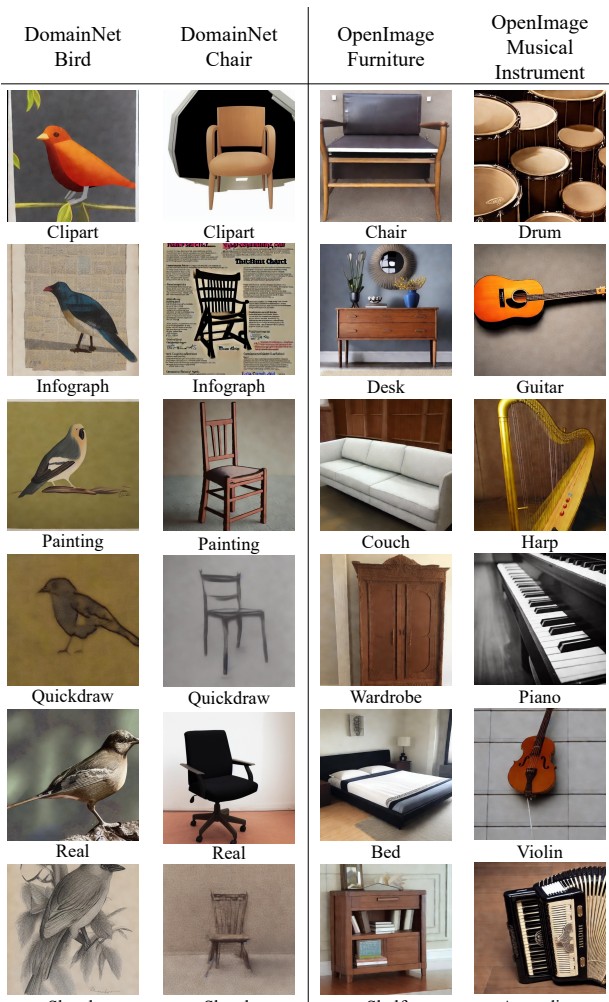

**Figure 2: The visualization of generated samples on Domain-Net and OpenImage.**

The visualization experiments in Figure 2, Figure 3, and other visual experiments in the supplementary materials clearly demonstrate that FedDEO is capable of generating synthetic dataset that complies with different client distributions when there are differences in style, subcategory, or background among clients, while being semantically correct. The synthetic datasets exhibit high quality and diversity, which is comparable to the client local datasets, underscoring the superior performance of FedDEO.

## 4.3 Ablation Experiments

To further demonstrate the performance of the proposed method, we conduct sufficient ablation experiments, thoroughly discussing the impacts of hyperparameters and other settings in our method, including the number of images in the synthetic dataset, the number of training epochs for the local descriptions, the used pre-trained DMs, the number of the clients, and more. Due to space limitations, we present partial experimental results and discussions here.

| | Common NICO++ | | | | | | | Unique NICO++ | | | | | | |
|---|---|---|---|---|---|---|---|---|---|---|---|---|---|---|
| | client0 | client1 | client2 | client3 | client4 | client5 | average | client0 | client1 | client2 | client3 | client4 | client5 | average |
| *Ceiling* | *50.24* | *54.36* | *63.35* | *64.82* | *61.99* | *65.09* | *59.98* | *74.02* | *78.9* | *79.68* | *74.47* | *77.34* | *77.47* | *76.98* |
| FedAvg | 18.23 | 27.79 | 36.32 | 52.42 | 37.96 | 39.24 | 35.32 | 34.96 | 58.98 | 38.41 | 63.41 | 45.44 | 59.76 | 50.16 |
| FedDF | 31.40 | 32.22 | 43.73 | 45.19 | 36.01 | 43.08 | 38.60 | 51.85 | 52.34 | 55.85 | 52.47 | 54.42 | 59.24 | 54.36 |
| FedProx | 37.31 | 35.95 | 42.78 | 48.92 | 41.07 | 47.53 | 42.26 | 54.55 | 60.51 | 54.05 | 58.34 | 55.69 | 57.78 | 56.82 |
| FedDyn | 36.83 | 37.85 | 45.21 | 51.38 | 42.74 | 44.36 | 43.06 | 55.29 | 59.71 | 56.68 | 61.74 | 48.99 | 61.31 | 57.28 |
| Prompts Only | 38.64 | 45.55 | 53.08 | 54.72 | 50.19 | 59.91 | 50.34 | 67.38 | 71.88 | 67.70 | 64.19 | 63.41 | 63.28 | 66.30 |
| FedDISC | 50.75 | 51.64 | 60.79 | 58.33 | 55.41 | 57.28 | 55.70 | 71.89 | 73.20 | 70.51 | 70.02 | 75.62 | 69.82 | 71.84 |
| FGL | 45.34 | 51.41 | 60.44 | 59.65 | 58.87 | 62.33 | 56.34 | 69.51 | 74.59 | 71.36 | 69.41 | 69.65 | 71.42 | 70.99 |
| FedCADO | **58.98** | 46.53 | 60.93 | 57.45 | 53.92 | 54.32 | 55.35 | 73.30 | 71.48 | 68.97 | 69.71 | 72.91 | 65.49 | 70.31 |
| FedDEO | 53.69 | **56.46** | **66.32** | **66.57** | **62.26** | **70.81** | **62.68** | **76.58** | **80.42** | **81.19** | **75.75** | **80.38** | **78.94** | **78.87** |

**Table 2: The performances of the compared methods on OpenImage, DomainNet, and NICO++ under the non-IID label distribution skew, where the italicized texts represent the performance ceiling of centralized training used as a reference, and bold texts represent the best performance of the compared methods.**

For more ablation experiments and detailed experimental settings, please refer to the supplementary materials. **The Number of Images.** The number of images in the synthetic dataset is one of the key factors influencing the performance of our method. The total number of images in the dataset is determined by the number of images generated under the guidance of each local description, defined as $R$ in the method section. Due to time limitations, we conduct relevant ablation experiments on the first 90 categories of DomainNet, and the experimental results are presented in Table 3. From the table, we can observe that as $R$ increases, there is indeed a noticeable improvement in the performance of the trained aggregated model. Additionally, it is worth noticing that the rate of performance improvement does not significantly slow down when the number of images increases from 30 to 50, proving the diversity of the synthetic dataset.

**The Epochs for Training Descriptions.** The number of training epochs for local descriptions, defined as $S$ in the method section, directly influence the characteristics of the client distributions captured by the local descriptions, thereby influencing the quality of the synthetic dataset and the upper bound mentioned in Theorem 1. Due to time limitations, we conduct relevant ablation experiments on the first 90 categories of DomainNet, and the experimental results are presented in Table 4. From the table, it can be observed that with the increase of $S$, the performance of the aggregated model shows a stable improvement. Additionally, when the descriptions are trained for only one epoch, they barely learn the stylistic information of the client distribution, hence only showing good performance on the *real* domain. This indicates the crucial role of the local descriptions in guiding the server image generation.

## 4.4 Limitations and Discussions

To more comprehensively demonstrate both the practicality and the limitations of the proposed method, we conduct discussions regarding three issues here: communication costs, computation costs, and the privacy concerns.

**Communication Costs.** The communication costs include both upload communication and download communication. We discuss these two parts separately.

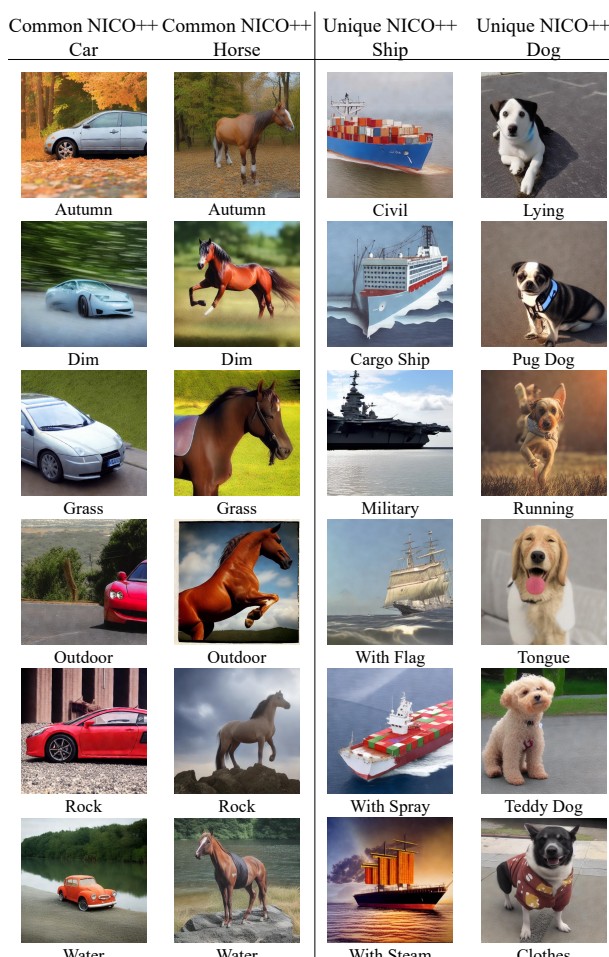

**Figure 3: The visualization of generated samples on NICO++.**

In the table 5, we demonstrate the upload communication costs for all compared methods. FedAvg, FedDF, FedProx, and FedDyn have similar communication costs, which are not repeated in the

| | DomainNet | | | | | | |
|---|---|---|---|---|---|---|---|
| | clipart | infograph | painting | quickdraw | real | sketch | average |
| R=10 | 55.16 | 21.95 | 45.61 | 12.64 | 67.41 | 39.32 | 40.43 |
| R=30 | 57.87 | 25.33 | 48.03 | 13.07 | 69.81 | 42.76 | 42.82 |
| R=50 | 60.64 | 27.85 | 50.96 | 13.62 | 71.75 | 45.81 | 45.11 |

**Table 3: The influence of the number of images.**

| | DomainNet | | | | | | |
|---|---|---|---|---|---|---|---|
| | clipart | infograph | painting | quickdraw | real | sketch | average |
| S=1 | 42.1 | 16.28 | 43.71 | 8.56 | 68.47 | 33.55 | 35.44 |
| S=10 | 57.87 | 25.33 | 48.03 | 13.07 | 69.81 | 42.76 | 42.82 |
| S=20 | 59.46 | 28.62 | 51.67 | 14.11 | 70.25 | 45.31 | 44.90 |

**Table 4: The influence of the epochs for training descriptions**

| Uploaded Parameters (M) | | | | |
|---|---|---|---|---|
| FedAvg | Ceiling | FedCADO | FedDISC | FedDEO |
| $20 \times 11.69 = 233.8$ | 270.95 | 11.69 | 4.23 | **3.54** |

**Table 5: Comparison about the communication costs.**

table. Prompts Only does not involve any communication costs, so it is not included. FGL is also not included in the comparison because the length of the text prompts generated in FGL is highly random. From table 5, It can be observed that FedDEO, with the least upload communication costs, is capable of training the aggregated model with better performance. This indicates the locally trained descriptions can more accurately convey client knowledge and efficiently guide the image generation on the server.

Regarding the download communication, it is undeniable that FedDEO does not have a significant advantage due to the need for clients to download DM to train local descriptions. However, we must point out that, on one hand, most FL methods based on base models, including diffusion-based OSFL methods such as Fed-DISC, FGL, and various federated fine-tuning methods [27, 36, 40], require downloading the base model to the clients, resulting in additional download communication. On the other hand, some existing works [17] focus on the use of DM on mobile devices. FedDEO is fully compatible with these methods and can effectively reduce the download communication. Therefore, the download communication of FedDEO is entirely acceptable.

**Computation Costs.** The computation costs include the computation costs on the client and the server. Since FedDEO fixes all parameters of the DM and only trains the local descriptions, the client computation costs of FedDEO is comparable to various federated fine-tuning methods [27, 36, 40], making it entirely acceptable. Additionally, with the same number of generated images, the computation cost on the server is identical to other diffusion-based OSFL methods. Therefore, although the computation cost poses some limitations, it does not significantly compromise the practicality of our method.

**Privacy Concerns.** Essentially, the local descriptions trained on the clients is a kind of model parameters, which is widely used in FL methods. Additionally, due to the low upload communication of Fed-DEO, where privacy leakage mainly occurs, FedDEO exhibits lower risk compared to other FL methods. To further validate FedDEO's

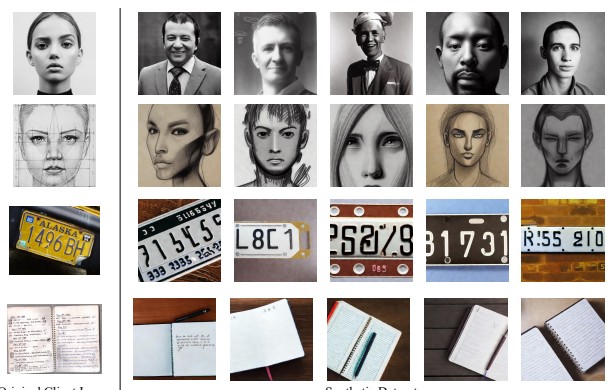

Original Client Image              Synthetic Dataset

**Figure 4: The visualization of privacy-sensitive information-related categories.**

performance in privacy protection, we conducted sufficient quantitative and visual experiments. We select some categories from OpenImage and DomainNet that may contain privacy-sensitive information, such as faces, license plates, books, etc. We train descriptions separately on images of these categories and generate synthetic datasets. The visualization results are shown in Figure 4. It can be observed that the synthetic datasets only share similar styles and identical semantics with the original client datasets. It is almost impossible to extract specific privacy-sensitive information from the descriptions, which aiming to characterize the overall distribution. Furthermore, we conduct more quantitation and visualization experiments, including overall comparisons between the synthetic dataset and client local datasets, as well as whether encryption of descriptions can be achieved through noise addition to the uploaded descriptions, etc. Due to space limitations, please refer to the supplementary materials for more detailed experimental settings and results about privacy issues.

## 5 CONCLUSIONS

In this paper, we propose FedDEO, which employs local descriptions trained on the clients as the medium to transfer distributed client knowledge to the server. Utilizing the powerful DM, the descriptions serves as conditions in generating the synthetic datasets that compiles with various client distributions, enabling the training of aggregated model. Theoretical analyses prove that the local descriptions can efficiently reduce the upper bound of KL divergence between the synthetic datasets and the client datasets, providing the theoretical foundation for other diffusion-based OSFL methods. Sufficient quantitation and visualization experiments on three large-scale real-world datasets demonstrate the performance of FedDEO. With advantages in communication and privacy protection, the trained aggregated model outperforms all compared methods and has the potential to outperform the performance ceiling of centralized training. As a novel exploration in diffusion-based OSFL, FedDEO further elucidates the significant potential of utilizing diffusion models and other foundation models in federated learning.

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
