# OpenReview forum: "FedDEO: Description-Enhanced One-Shot Federated Learning with Diffusion Models"
_acmmm.org/ACMMM/2024/Conference — MM2024 Poster_

### Official Review · Reviewer_y9Qh · 2024-05-13

**Rating:** 3
**Confidence:** 3

**Summary:**

In this paper, the authors propose FedDEO, a Description-Enhanced One-Shot Federated Learning Method with DMs, offering a novel exploration of utilizing the DM in OSFL. The core idea of their method involves training local descriptions on the clients, serving as the medium to transfer the knowledge of the distributed clients to the server.

**Strengths:**

The authors applied DM to federated learning and achieved better performance than the centralized training.

**Limitations:**

1. I think the experimental comparison is unfair. The authors claim that they used a very adequately pre-trained diffusion model, but that would lead to the introduction of very much additional knowledge. It can be assumed that the performance improvement is not due to the federated paradigm proposed by the authors.
2. I think the authors' proposed method benefits from existing methods, such as DM, and does not contribute usefully to the federated learning community.

**Suitability:**

2

---

### Official Review · Reviewer_89ZF · 2024-05-23

**Rating:** 5
**Confidence:** 4

**Summary:**

This manuscript proposes FedDEO, a description-enhanced one-shot federated learning (OSFL) method leveraging diffusion models (DMs). This method addresses the limitations of traditional OSFL methods, which often require public datasets or uniform feature extractors, by introducing local descriptions as a new medium for knowledge transfer. These descriptions guide the generation of synthetic datasets that adhere to the clients' data distributions, improving model performance, communication efficiency, and privacy protection. Experiments have been conducted on three datasets.

**Strengths:**

1. The idea of using local descriptions to guide the generation of synthetic datasets is novel and makes sense.
2. Theoretical analyses have been thoroughly conducted, demonstrating the bounded KL divergence between synthetic and client data distributions. The quality and diversity of the synthetic datasets are comparable with the original client datasets.
3. The manuscript is generally well-written and organized, making the complex methodology understandable. Many experiments and method details are demonstrated in the main text and supplementary materials.

**Limitations:**

1. The proposed FedDEO outperforms even the performance ceiling of centralized training. Please provide an analysis of whether it is overfitting.
2. Diffusion-based OSFL baseline methods used in this manuscript, such as FedDISC, FGL, and FedCADO, are still in the preprint stage on arXiv and have not been formally published. To enhance the credibility and robustness of the evaluations, the manuscript could benefit from including comparisons with more established and formally published baselines in diffusion-based federated learning. This would provide a more comprehensive and reliable assessment of the proposed method's performance.

**Suitability:**

2

---

### Official Review · Reviewer_MRey · 2024-05-26

**Rating:** 4
**Confidence:** 3

**Summary:**

This paper proposed a new one-shot federated learning by leveraging the pre-trained diffusion model. It involves a local descriptions
training step to capture the characteristics of client distributions, and a data synthesis step that utilizes the local descriptions in the server. Experiments demonstrate that the proposed description-enhanced generation can promote the model performance significantly.

**Strengths:**

1. The proposed framework is simple yet effecitive, non-trivil improvement can be obtained compared with other baselines
2. Theoretical and experimental results seem to be promising

**Limitations:**

1. the data synthesis based approaches highly rely on the assumption that the generative model have been trained on the data with the same or similar semantic meaning, which may not be versatile enough for all kinds of FL tasks.
2. If the pre-trained diffusion models could be utilized, why not compare those methods that leverage the well pre-trained models on large scale data, such as ResNet and ViT with PEFT techniques.
3. Will the style/domain of image a kind of sensitive information of local client?
4. While the diffusion model could generate natural images/objects, what if the target classification tasks are not natural images/objects

**Suitability:**

2

---

### Meta-Review · Area_Chair_v6fs · 2024-06-30

**Recommendation:** Accept (Poster)
**Confidence:** 4

**Metareview:**

FedDEO is a description-enhanced one-shot federated learning method leveraging diffusion models. The technique uses local descriptions to guide the generation of synthetic datasets, improving model performance, communication efficiency, as well as privacy protection.

Reviewers noted several strengths, including the novel idea of using local descriptions for synthetic data generation, thorough theoretical analysis, and comprehensive experiments. However, concerns were raised about the fairness of experimental comparisons, potential overfitting, and the contribution of the method to the federated learning community.

Addressing these concerns and providing more detailed comparisons and evaluations could significantly enhance the paper's impact.